# Attention Contrastive Decoding: Preserving Coherence While Mitigating Hallucinations in Large Vision-Language Models

## Abstract

Large Vision-Language Models (LVLMs) exhibit remarkable multimodal capabilities but frequently produce factually inconsistent hallucinations. While Contrastive Decoding (CD) methods offer a training-free approach to hallucination mitigation, they operate at the logits level, compromising output coherence and diversity. Through systematic analysis, we show that logits-level subtraction disrupts intrinsic language generation mechanisms, requiring restrictive penalty mechanisms that further limit diversity. We propose Attention Contrastive Decoding (ACD), which transfers contrastive operations to the attention layer and employs an Adaptive Subtraction Strategy (ASS) to identify and suppress hallucination-prone attention patterns. Experiments demonstrate that ACD generates more coherent content with significantly reduced hallucinations without requiring penalty mechanisms, effectively leveraging the inherent continuity of attention mechanisms to advance reliable multimodal generation. Code is available at `https://anonymous.4open.science/r/ACD-00C6`.

## 1 INTRODUCTION

In recent years, the exponential growth of computational capabilities coupled with the rapid expansion of multimodal datasets has catalyzed the emergence of Large Vision-Language Models (LVLMs) Liu et al. (2023b;c; 2024a); Zhu et al. (2023); Bai et al. (2023); Li et al. (2023a); Driess et al. (2023); Achiam et al. (2023); Chowdhery et al. (2023); Alayrac et al. (2022). These sophisticated architectures integrate meticulously designed visual encoders with large language models (LLMs), effectively extending the robust linguistic comprehension and generation capabilities of LLMs into the domain of multimodal interaction. Such models have demonstrated exceptional performance across image captioning Lin et al. (2014); Li et al. (2023a), visual question answering Liu et al. (2023c;b; 2024a); Hudson & Manning (2019), and complex cross-modal reasoning tasks Lu et al. (2022); Alayrac et al. (2022); Achiam et al. (2023), thereby establishing a solid technical foundation for more natural and diversified human-machine interaction paradigms.

Despite significant advances in LVLMs, a critical challenge persists—the "hallucination" phenomenon Li et al. (2023b); Fu et al. (2023b); Yue et al. (2024). These hallucinations manifest as textual outputs that, while grammatically and semantically coherent, present factual inconsistencies with the input visual content. Unlike hallucinations in pure text-based LLMs, LVLM hallucinations exhibit distinct cross-modal characteristics that not only compromise textual quality but also involve profound semantic inconsistencies between visual and linguistic modalities. Specifically, LVLM hallucinations typically manifest in three characteristic patterns: (1) generation of responses entirely disconnected from the image content, indicating complete modal disassociation; (2) erroneous identification of non-existent visual elements (such as colors, quantities, or spatial relationships), demonstrating flawed visual perception; and (3) inaccurate abstract summarization of visual content, reflecting semantic reasoning biases. These cross-modal inconsistencies substantially undermine model credibility in practical applications and pose significant safety risks in critical domains such as medical diagnostics and autonomous driving.

Alleviating hallucinations in LVLMs has emerged as a critical research focus, with training-free Contrastive Decoding (CD) strategies Li et al. (2025); Huo et al. (2024); Liu et al. (2024c); An

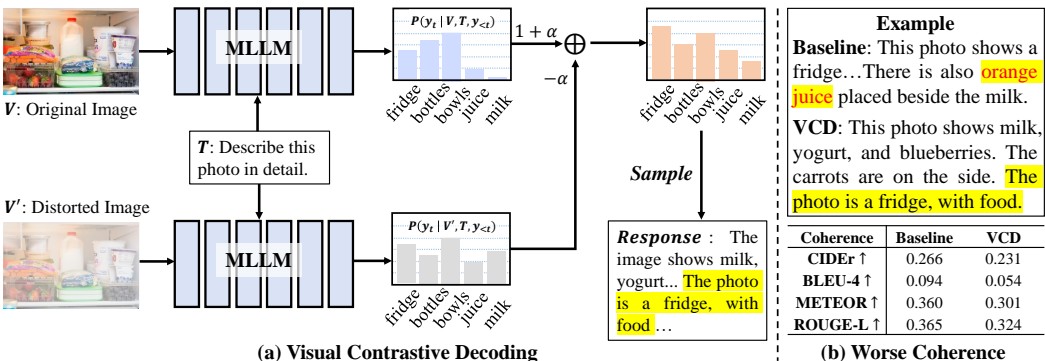

(a) Visual Contrastive Decoding

(b) Worse Coherence

Figure 1: ((a) Visual Contrastive Decoding (VCD). Comparing the output distributions from the original and distorted inputs. (b) While VCD mitigates hallucination generation, its discrete logits-level contrastive adjustments adversely affect generation quality, resulting in compromised coherence and fluidity in the produced content.

et al. (2025); Leng et al. (2024); Wang et al. (2024c); Chen et al. (2024) representing a prominent paradigm. These approaches mitigate hallucinations by differentially adjusting token generation probability distributions through two principal mechanisms: hallucination induction and probability recalibration. In the hallucination induction framework, several methodological variants have been proposed. Visual Contrastive Decoding (VCD) Leng et al. (2024) artificially amplifies the model's reliance on linguistic priors by either introducing Gaussian noise into visual inputs or completely eliminating visual signals Liu et al. (2024c), thereby constructing a hallucination-rich probability distribution. Similarly, Instructional Contrastive Decoding (ICD) Wang et al. (2024c) employs adversarial prompts (e.g., "you are a confused object detector") to elicit hallucinated outputs. Self-Introspective Decoding (SID) Huo et al. (2024) implements strategic masking of high-attention visual regions during the decoding process, effectively exposing vulnerable areas prone to hallucination generation. The probability recalibration mechanism precisely subtracts this artificially constructed hallucination distribution from the original generation probability, as illustrated in Figure 1 (a), effectively suppressing the production of factually inconsistent information and enhancing the model's fidelity to visual evidence. This parameter-update-free contrastive mechanism offers a convenient and theoretically interpretable solution for hallucination mitigation in LVLMs, significantly enhancing the accuracy and reliability of generated content while preserving the model's inherent capabilities.

Despite substantial empirical evidence Leng et al. (2024); Wang et al. (2024c); Huo et al. (2024) demonstrating the efficacy of CD strategies in mitigating LVLM hallucinations, our investigation reveals that CD approaches significantly compromise generation coherence (manifested as invalid token outputs) under specific conditions. The fundamental limitation lies in CD's crude direct subtraction operation at the logits layer, which lacks nuanced regulation of the natural language generation process. Specifically, due to the inherent discontinuity and volatility of logits distributions, **logit-level subtraction forcibly alters vocabulary distribution probabilities, thereby disrupting the model's intrinsic language generation mechanisms**. This direct intervention inadequately accounts for contextual dependencies and logical coherence in language generation, resulting in semantic and structural inconsistencies in generated outputs, as illustrated in Figure 1 (b). Our systematic experimentation with the LLaVA 1.5-7B Liu et al. (2023b) model on the CHAIR dataset Yue et al. (2024), demonstrates that VCD consistently underperforms across multiple evaluation metrics: CIDEr (measuring description quality), METEOR (assessing semantic comprehension accuracy), ROUGE-L (quantifying information completeness), and BLEU (evaluating generation precision) scores significantly trail those of conventional decoding methods. These empirical findings explicitly reveal an inherent contradiction in current contrastive decoding paradigms-while logits-level contrast operations effectively suppress hallucinations, they inevitably degrade linguistic quality in model outputs, presenting a critical trade-off challenge for LVLM deployment. Concurrently, probability recalibration mechanisms at the logits level frequently amplify tail token probabilities due to distribution discontinuities, resulting in incoherent generations. To mitigate this issue, these strategies typically incorporate penalty mechanisms that filter low-probability tokens. However, our in-depth analysis indicates that this compensatory mechanism not only fails to adequately resolve the issue but raises additional challenges. **First, penalty mechanisms that mask low-probability**

**tokens sacrifice decoding diversity** (constraining the model to high-probability tokens exclusively). This implies that penalty mechanisms function primarily to counteract CD's adverse effects rather than addressing the hallucination problem directly. This compensatory processing significantly restricts the richness and creativity of generated content, biasing outputs toward high-frequency, conventional expressions lacking natural linguistic variation and innovation. **Second, the intensity calibration of current penalty mechanisms predominantly relies on heuristic algorithms** (utilizing manually defined thresholds), severely limiting their adaptability and inevitably resulting in invalid token outputs. In diverse linguistic contexts and tasks, fixed penalty strategies inadequately accommodate variable generation requirements and cannot dynamically adjust filtering intensity based on contextual semantics, further exacerbating the inconsistency and instability of generated outputs.

To address the limitations of conventional CD methods, we propose an innovative decoding strategy—Attention Contrastive Decoding (ACD). Unlike traditional CD approaches that perform crude direct subtraction at the logits level, ACD elegantly transposes the contrastive mechanism to the attention layer while incorporating an Adaptive Subtraction Strategy (ASS), achieving more refined hallucination control. The cornerstone of ASS lies in its comparative mechanism: by contrasting attention-layer response intensities between original inputs and hallucination-inducing inputs, it precisely identifies potential hallucination-triggering regions. Based on this assessment, ASS adaptively suppresses the model's attention allocation to hallucination-prone areas, specifically inhibiting hallucination generation at its source, thereby producing more coherent content. This fine-grained control mechanism enables the model to maintain linguistic quality while preserving visual information accuracy. ACD inherently benefits from operating at the transformer's attention layer. This architectural advantage stems from the attention mechanism's capacity to weight input information and adjust according to contextual cues, facilitating smooth information flow Vaswani et al. (2017); Castin et al. (2023). Consequently, attention-based ACD naturally generates more coherent outputs, thus circumventing the need for penalty terms while preserving generation diversity. This adaptive contrastive mechanism at the attention level not only precisely localizes and suppresses hallucination sources but also preserves the model's intrinsic language generation capabilities, offering a more elegant and effective solution for hallucination mitigation in LVLMs.

In summary, our main contributions are three-fold: (1) We establish that logits-layer subtraction in Contrastive Decoding fundamentally compromises generation quality, revealing how penalty mechanisms fail to resolve invalid token generation while simultaneously restricting output diversity. (2) We propose Attention Contrastive Decoding (ACD), which relocates contrastive operations from logits to attention layers, harnessing the inherent continuity of attention mechanisms. Our Adaptive Subtraction Strategy (ASS) precisely identifies hallucination-prone attention patterns through differential visual response analysis. (3) Our empirical evaluation demonstrates ACD's capacity to generate more coherent, higher-quality content with significantly reduced hallucinations without penalty mechanisms, advancing the state-of-the-art in reliable vision-language model decoding.

## 2 RELATED WORK

### 2.1 HALLUCINATION IN LVLMS

Research addressing hallucinations in large vision-language models (LVLMs) can be categorized into three primary approaches. First, contrastive decoding techniques identify and suppress hallucinated content without parameter updates. Representative methods include VCD Leng et al. (2024), which contrasts outputs from standard and distorted inputs; ICD Wang et al. (2024c), which leverages instruction perturbations; and SID Huo et al. (2024), which strategically removes attention patches. Advanced variants such as HALC Chen et al. (2024), VaLiD Wang et al. (2024b), CMVED Li et al. (2025), PAI Liu et al. (2024c), and AGLA An et al. (2025) further refine these techniques through targeted attention manipulation and visual component modification. Second, fine-tuning and optimization strategies employ curated datasets to mitigate hallucinations. These approaches incorporate negative data Liu et al. (2023a), counterfactual data Yu et al. (2024), and dataset purification Wang et al. (2024c). Frameworks including HALVA Sarkar et al. (2024), HACL Jiang et al. (2024a), PerturboLLaVA Chen et al. (2025), and PATCH Shang et al. (2024) implement contrastive learning and feature integration, while HIO Lyu et al. (2024), Octopus Suo et al. (2025), OPA Yang et al. (2025b), TL-DPO Yoon et al. (2025), and VASparse Zhuang et al. (2025) utilize reinforcement learning and attention sparsification to address specific hallucination patterns. Third,

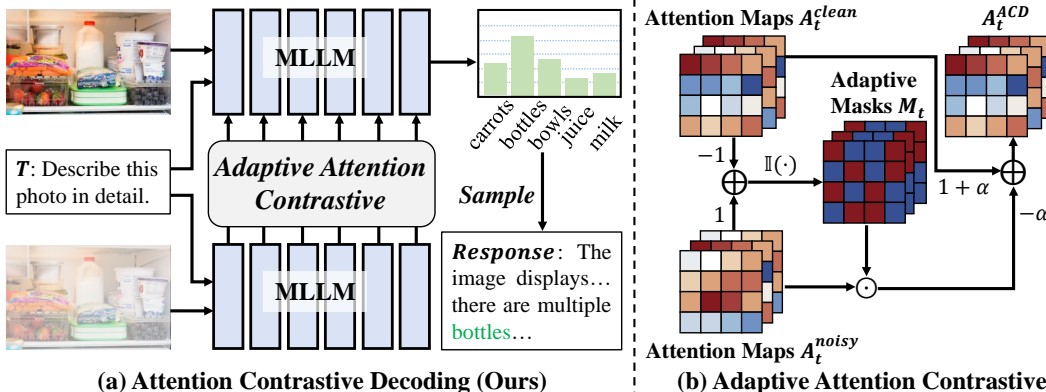

**(a) Attention Contrastive Decoding (Ours)** | **(b) Adaptive Attention Contrastive**

Figure 2: **Overview of our method.** (a) The ACD approach ingeniously transfers contrastive adjustments from the discrete logits layer to the continuous attention layer, achieving superior hallucination mitigation while enhancing the coherence of generated content. (b) Concurrent with Adaptive Attention Contrastive Decoding, we introduce Attention Subtraction Strategy (ASS). The ASS strategy precisely identifies potential hallucination-prone regions by comparing attention response intensities between the original input and hallucination-inducing input, thereby selectively suppressing the sources of hallucination generation at their origin.

methods focusing on visual feature representation enhancement include DeCo Wang et al. (2024a), VTI Liu et al. (2024b), and VDGD Ghosh et al. (2024), which enrich feature detail and stability. Additionally, ProjectAway Jiang et al. (2024b), TAME Tang et al. (2025a), OPERA Huang et al. (2024), VAR Jiang et al. (2025), FarSight Tang et al. (2025b), Nullu Yang et al. (2025a), and ClearSight Yin et al. (2025) manipulate attention mechanisms and feature spaces to suppress hallucinations through orthogonal projection, dynamic modification, and guided visual processing.

## 2.2 CONTRASTIVE DECODING IN HALLUCINATION

Contrastive Decoding (CD) methods provide an elegant, training-free approach to mitigating hallucinations in LVLMs without parameter optimization. These strategies reduce hallucination generation by contrastively adjusting token probability distributions through two key mechanisms: hallucination induction and probability adjustment. Representative works include VCD Leng et al. (2024), which injects Gaussian noise into visual inputs to construct hallucination-rich distributions; ICD Wang et al. (2024c), which employs negative prompts to induce hallucinations; and SID Huo et al. (2024), which strategically masks high-attention regions to expose model vulnerabilities. Advanced variants include HALC Chen et al. (2024) with automatic hallucination correction, VaLiD Wang et al. (2024b) comparing early-layer outputs, CMVED Li et al. (2025) masking cross-modal attention, PAI Liu et al. (2024c) removing visual components entirely, and AGLA An et al. (2025) emphasizing critical regions through image-text matching models. Despite demonstrated efficacy in hallucination mitigation, conventional CD strategies operate exclusively at the logits distribution level, substantially compromising output coherence and diminishing generative diversity.

## 3 METHOD

### 3.1 PRELIMINARY

**Vanilla Decoding.** We formalize a general LVLM, denoted as $\theta$, comprised of three principal components: a vision encoder, a vision-text interface, and a large language model (LLM) decoder. The operational pipeline begins with the vision encoder processing an input image $v$ to extract visual embeddings. These embeddings are subsequently transformed by the vision-text interface (e.g., linear projection matrices Liu et al. (2023c;b) or Q-former Bai et al. (2023); Li et al. (2023a)) to achieve modality alignment with the textual query $x$. The aligned multimodal representation is then fed to the LLM decoder, which autoregressively generates textual output $y$ according to:

$$y_t \sim p_\theta(y_t|v, x, y_{<t}), \quad p_\theta(y_t|v, x, y_{<t}) \propto \exp\left(\text{logit}_\theta(y_t|v, x, y_{<t})\right), \quad (1)$$

where, $y_t$ represents the $t$-th token of $y$, while $y_{<t}$ refers to the sequence of tokens generated prior to $t$-th step. The function $\text{logit}_\theta$ is the logit distribution function. The attention mechanism within each decoder head is formulated as:

$$\text{Attention}(Q, K) = \text{Softmax}\left(\frac{QK^\top}{\sqrt{d}}\right), \qquad (2)$$

where, $Q \in \mathbb{R}^{L \times D}$ and $K \in \mathbb{R}^{L \times D}$ are the query matrix and the key matrix, respectively, $D$ represents the dimension, and $L$ denotes the length of the sequence. During inference, to optimize computational efficiency, the model maintains a key-value cache storing $K$ and $V$ from previous decoding steps, thereby avoiding redundant calculations. Consequently, the attention computation for the $t$-th decoding step is expressed as:

$$\text{Attention}(q_t, K_{\leq t}) = \text{Softmax}\left(\frac{q_t K_{\leq t}^\top}{\sqrt{d}}\right), \qquad (3)$$

where $q_t$ is the query for the current decoding step, and $K_{\leq t}$ represents the keys up to and including step $t$.

**Contrastive Decoding.** CD mitigates hallucination through differential adjustment of token generation probabilities, employing two fundamental mechanisms: hallucination induction and probability recalibration. Within the hallucination induction framework, several methodological variants have emerged: VCD Leng et al. (2024) induces controlled hallucinations by either introducing Gaussian noise into visual inputs or completely eliminating visual signals Liu et al. (2024c) (denoted as $v'$); ICD Wang et al. (2024c) constructs adversarial prompts ($x'$) to elicit hallucinations; while SID Huo et al. (2024) implements strategic masking of high-attention visual regions during the decoding process (also representable as $v'$). Subsequently, a new contrastive probability distribution is computed by leveraging the differences between these two distributions. This contrastive distribution, denoted as $p_{cd}$, is defined as:

$$p_{cd}(y \mid v, v', x') = \text{softmax}\left[(1 + \alpha)\text{logit}_\theta(y \mid v, x) - \alpha \text{logit}_\theta(y \mid v', x')\right], \qquad (4)$$

where larger $\alpha$ values indicate a stronger amplification of differences between the two distributions ($\alpha = 0$ degenerates to Vanilla decoding). A fundamental limitation in Equation 4 is its uniform penalization mechanism that indiscriminately penalizes all outputs from distorted inputs, disregarding their potential linguistic validity and reasoning coherence. This approach risks suppressing legitimate outputs while potentially promoting implausible generations. Consequently, effective CD frameworks require implementing an plausibility constraint that dynamically calibrates penalization intensity based on confidence metrics derived from original input distributions.

$$\mathcal{V}_{\text{head}}(y_{<t}) = \{y_t \in \mathcal{V} : p_\theta(y_t \mid v, x, y_{<t}) \geq \beta \max_w p_\theta(w \mid v, x, y_{<t})\},$$
$$p_{cd}(y_t \mid v, v', x') = 0, \text{ if } y_t \notin \mathcal{V}_{\text{head}}(y_{<t}), \qquad (5)$$

where $\mathcal{V}$ is the output vocabulary of LVLMs and $\beta$ is a hyperparameter in $[0, 1]$ for controlling the truncation of the next token distribution. Larger $\beta$ indicates more aggressive truncation, keeping only high-probability tokens. Combining the CD and the plausibility constraint, we obtain the full formulation:

$$y_t \sim \text{softmax}\left[(1 + \alpha)\text{logit}_\theta(y_t \mid v, x, y_{<t}) - \alpha \text{logit}_\theta(y_t \mid v', x', y_{<t})\right],$$
$$\text{subject to} \quad y_t \in \mathcal{V}_{\text{head}}(y_{<t}). \qquad (6)$$

## 3.2 ATTENTION CONTRASTIVE DECODING (ACD)

Traditional CD methods, as formulated in Equation 6, mitigate hallucinations through direct subtraction operations at the logits layer. However, this approach operating in discontinuous and volatile probability spaces significantly compromises language generation quality. To address this fundamental limitation, ACD strategically transposes the contrastive mechanism from the logits distribution layer to the attention layer within the Transformer architecture of the LVLM decoder, with the comprehensive methodological framework illustrated in Figure 2. Specifically, during the $t$-th

decoding step, the model computes parallel attention distributions under two distinct input configurations. For the original visual input and hallucination-inducing input conditions, the respective attention distributions are formally defined as:

$$A_t^{clean} = \text{Softmax}\left(\frac{q_t K_{\leq t}^{clean\top}}{\sqrt{d}}\right), \quad A_t^{noisy} = \text{Softmax}\left(\frac{q_t K_{\leq t}^{noisy\top}}{\sqrt{d}}\right), \tag{7}$$

where $q_t$ denotes the query vector for the current decoding step, and $K_{\leq t}^{clean\top}$ and $K_{\leq t}^{noisy\top}$ represent the key matrix caches computed using original inputs and hallucination-inducing inputs, respectively. In contrast to traditional CD methods that perform direct subtraction operations at the logits layer, ACD applies the contrastive mechanism at the attention layer:

$$A_t^{ACD} = (1 + \alpha)A_t^{clean} - \alpha \, ASS(A_t^{clean}, A_t^{noisy}), \tag{8}$$

where $ASS(\cdot)$ represents our proposed Adaptive Subtraction Strategy, designed to precisely identify and modulate attention distributions potentially conducive to hallucination generation.

### 3.3 ADAPTIVE SUBTRACTION STRATEGY (ASS)

The Adaptive Subtraction Strategy (ASS) constitutes the principal innovation of our ACD methodology. This mechanism functions by comparative analysis of attention response intensities between original visual inputs and hallucination-inducing inputs, thereby precisely identifying potential hallucination-triggering regions. Based on this sophisticated assessment, ASS adaptively suppresses the model's attention allocation to hallucination-prone areas, effectively inhibiting hallucination generation at its source. Specifically, the ASS mechanism is implemented through the following procedure:

$$M_t = \mathbb{I}(A_t^{clean} - A_t^{noisy} < 0), \quad A_t^{ACD} = (1 + \alpha)A_t^{clean} - \alpha A_t^{noisy} \odot M_t, \tag{9}$$

where $\mathbb{I}(\cdot)$ denotes the indicator function, returning 1 when the condition is satisfied and 0 otherwise; $\odot$ represents element-wise multiplication. The theoretical foundation of this masking mechanism is as follows:

- When $A_t^{clean} - A_t^{noisy} > 0$, it indicates that in hallucination-inducing inputs, the model's attention allocation undergoes a shift, resulting in semantic information loss in the corresponding region. In such instances, we preserve the original attention distribution without applying subtraction operations to maintain semantic integrity.
- When $A_t^{clean} - A_t^{noisy} < 0$, it signifies that the model allocates disproportionate attention to incorrect regions in hallucination-inducing inputs, potentially precipitating hallucination generation. In these cases, we implement subtraction operations to suppress these hallucination-prone attention allocations.

Unlike traditional CD methodologies that necessitate the application of penalty mechanisms as formulated in Equation 5, ACD inherently generates more coherent outputs due to its intrinsic distributional smoothness at the attention layer, thus eliminating the requirement for additional penalty terms while preserving generation diversity. The complete decoding procedure is formally detailed in Algorithm C.

## 4 EXPERIMENTS

### 4.1 EXPERIMENTAL SETTINGS

Following established evaluation protocols from prior work Leng et al. (2024); Huo et al. (2024); An et al. (2025), we assess our method's efficacy across standard benchmarks. Comprehensive details regarding datasets and evaluation metrics are provided in Appendix D.

**Benchmarks. POPE** Li et al. (2023b): A binary classification framework (20,000+ QA pairs) assessing object hallucination through yes/no questions with random, popular, and adversarial sampling strategies. Performance measured via Accuracy, Precision, Recall, and F1. **MME** Fu et al.

(2023a): Comprehensive evaluation across 14 subtasks spanning perception (object existence, color, position) and cognition (reasoning, computation). Each image features complementary yes/no questions evaluated through accuracy metrics. **CHAIR** Yue et al. (2024): Quantifies hallucination using image annotations rather than lexical similarity. Primary metrics include CHAIRs (sentence-level), CHAIRi (instance-level), and recall for measuring semantic comprehensiveness. **LLaVA-Bench (In-the-Wild)** Liu et al. (2023b): Evaluates generalization through 24 challenging images (memes, paintings, sketches) with 60 questions. Performance assessed by GPT-5 OpenAI (2025) on factual accuracy and descriptive richness.

**Implementation Details.** We evaluate the effectiveness of our proposed ACD on several established LVLMs, including LLaVA-1.5 Liu et al. (2023b) (7B and 13B) with MLP projection layers and Qwen-VL (7B) Bai et al. (2023) with q-former projection layers. Inference experiments for LLaVA-1.5-7B and Qwen-VL(7B) were conducted on a single NVIDIA RTX 3090 GPU, while LLaVA-1.5-13B was evaluated on a single NVIDIA A6000 GPU. The hyperparameter $\alpha$ was set to 0.15 throughout all experiments. To demonstrate the efficacy of our approach, we compare ACD against classical CD strategies designed for hallucination mitigation in LVLMs, including VCD Leng et al. (2024), ICD Wang et al. (2024c), and SID Huo et al. (2024).We further incorporate our Attention Contrastive Decoding methodology into established contrastive frameworks, evaluating its efficacy across diverse hallucination-prone scenarios while eliminating dependency on restrictive penalty mechanisms. Throughout all experiments, we maintain consistent configuration parameters and employ sampling as the default decoding strategy.

### 4.2 EXPERIMENTAL RESULTS

**Experiments on CHAIR.** Unlike the binary response paradigms (yes/no) characteristic of POPE and MME evaluations, the CHAIR dataset presents a substantially more challenging benchmark requiring detailed descriptive generation, which inherently increases susceptibility to hallucination phenomena. As demonstrated in Table 1, our proposed methodology consistently improves performance across various contrastive decoding-induced hallucination metrics. Specifically, ACD effectively reduces object hallucinations in generated captions, as evidenced by lower CHAIRS and CHAIRI scores. Concurrently, ACD enhances the detailedness of the generated captions, as indicated by higher Recall scores. These results demonstrate that ACD achieves an optimal balance between factual accuracy and descriptive richness in open-ended caption generation. This performance advantage derives from the attention-layer contrastive adjustment mechanism and ASS introduced in Sections 3.2 and 3.3, which collectively enable more precise visual-linguistic alignment while preserving narrative coherence.

**Experiments on LLaVA-Bench-Wild.** Following established protocols in prior research Leng et al. (2024); Huo et al. (2024); An et al. (2025), we employed a strong LVLM for evaluation, specifically utilizing SOTA GPT-5 OpenAI (2025) as an independent assessor to evaluate both accuracy and detail comprehensiveness. As demonstrated in Table 2, our proposed ACD methodology consistently outperforms the VCD approach across all evaluation dimensions. For LLaVA-1.5, ACD achieved accuracy improvements of 1.59 points and detail enhancements of 0.65 points; similarly, for Qwen-VL, we observed accuracy gains of 1.71 points and detail improvements of 0.96 points. Notably, these performance enhancements manifest across diverse task categories (Conversational, Detailed, and Complex), with the most substantial improvements consistently observed in the Detail category. This performance pattern suggests that ACD's attention-layer contrastive mechanism and adaptive subtraction strategy effectively enhance the model's capacity to attend to fine-grained visual information while simultaneously suppressing attention to non-existent or hallucinated regions, thereby significantly improving comprehensive perception capabilities. The empirical results confirm that smoothing contrastive adjustments at the attention layer rather than at logits enables more nuanced visual-linguistic alignment with greater fidelity to the actual visual content.

**Experiments on POPE.** We evaluated the integration of our ACD methodology with various hallucination-inducing scenarios, as presented in Table 4. Experimental results demonstrate that our approach achieves performance parity with conventional CD techniques while eliminating the necessity for auxiliary penalty term constraints, simultaneously producing more coherent textual outputs. Upon further analysis, we discovered that the POPE dataset's benchmark exhibits inherent limitations due to its binary (yes/no) question structure, which constrains the generation of nuanced, gradient responses. Consequently, while our algorithm does not demonstrate pronounced

Table 1: Results of CHAIR hallucination evaluation for the open-ended caption generation task.

| Model | LLaVA | | | Qwen-VL | | |
|---|---|---|---|---|---|---|
| Decoding | $Chair_s \downarrow$ | $Chair_i \downarrow$ | $Recall \uparrow$ | $Chair_s \downarrow$ | $Chair_i \downarrow$ | $Recall \uparrow$ |
| Regular | 54.1 | 18.5 | 73.4 | 50.5 | 15.0 | 71.1 |
| VCD | 51.8 | 16.2 | 76.8 | 47.5 | 13.7 | 71.5 |
| **+ACD** | **51.0** | **14.3** | **78.3** | **47.0** | **12.5** | **72.1** |
| ICD | 52.1 | 15.5 | 76.7 | 48.1 | 13.1 | 69.6 |
| **+ACD** | **50.2** | **13.9** | **78.5** | **46.5** | **12.6** | **71.9** |
| SID | 50.5 | 14.1 | 78.2 | 46.0 | 12.7 | 72.5 |
| **+ACD** | **49.6** | **13.3** | **79.4** | **45.0** | **11.1** | **73.6** |

Table 2: Results of GPT-5 evaluation on the LLaVA-Bench-Wild dataset.

| Model | Decoding | Conv | | Detail | | Complex | | Total | |
|---|---|---|---|---|---|---|---|---|---|
| | | Acc. | Detail | Acc. | Detail | Acc. | Detail | Acc. | Detail |
| LLaVA-1.5 | VCD | 4.67 | 3.67 | 4.50 | 5.67 | 3.88 | 4.88 | 4.30 | 4.75 |
| | **+ACD** | **6.83** | **3.71** | **7.50** | **7.33** | **3.97** | **5.21** | **5.89** | **5.40** |
| Qwen-VL | VCD | 5.61 | 3.91 | 5.55 | 5.53 | 5.13 | 4.98 | 5.40 | 4.85 |
| | **+ACD** | **7.47** | **4.58** | **8.20** | **7.56** | **6.02** | **5.42** | **7.11** | **5.81** |

Table 3: Results from Comprehensive Evaluation on the MME Benchmark.

| Method | Perception | Cognition |
|---|---|---|
| Regular | 1440.0 | 294.6 |
| VCD | 1475.4 | 284.3 |
| **+ACD** | **1497.7** | **298.5** |
| ICD | 1464.2 | 287.2 |
| **+ACD** | **1488.6** | **290.3** |
| SID | 1481.4 | 291.7 |
| **+ACD** | **1518.7** | **299.1** |

advantages on this particular dataset, it exhibits statistically significant performance improvements on more sophisticated tasks requiring elaborate response generation, such as CHAIR and LLaVA-Bench evaluations, thereby validating its efficacy in complex multimodal reasoning scenarios.

**Experiments on MME.** As evidenced in Table 3, our ACD method achieves superior performance compared to the original VCD across mean metrics for person perception and recognition tasks, demonstrating its efficacy in mitigating hallucinations while enhancing the general capabilities of LVLMs. Notably, despite the MME benchmark's binary evaluation protocol (yes/no without intermediate response options), its comprehensive structure—encompassing 14 distinct subtasks across perception domains (object existence, color attribution, spatial positioning) and cognition functions (logical reasoning, computational inference)—presents sufficiently challenging and diverse evaluation scenarios to effectively highlight the advantages of our proposed methodology. The breadth and complexity of these assessment criteria provide robust validation for our approach's superiority in maintaining factual consistency while preserving model versatility.

### 4.3 ABLATION STUDY AND ANALYSIS

**Coherence Analysis.** To rigorously evaluate ACD's coherence efficacy, we employed the CHAIR dataset—a challenging benchmark requiring detailed descriptive generation rather than the binary responses (yes/no) characteristic of POPE and MME evaluations. As illustrated in Table 5, the VCD method exhibits suboptimal performance across all generation quality assessment metrics relative to conventional decoding approaches, suggesting that its logits-level visual contrastive mechanism fundamentally compromises textual coherence. In contrast, our proposed ACD methodology not only preserves coherence levels comparable to conventional decoding but also demonstrates modest yet consistent enhancements in CIDEr, BLEU-4, METEOR, and ROUGE-L metrics. These empirical findings provide compelling evidence for ACD's dual capability: simultaneously mitigating hallucinations while maintaining—and in certain aspects, enhancing—language generation quality. This balanced performance establishes ACD's particular efficacy for high-fidelity multimodal generation tasks where both factual accuracy and linguistic coherence are essential requirements.

**Impact of Penalty Terms and ASS on CD.** Using the canonical Contrastive Decoding implementation,VCD, as our experimental baseline, Table 6 demonstrates that while CD methods achieve substantial hallucination mitigation compared to conventional decoding strategies, the removal of the adaptive penalty term results in significant performance degradation. This deterioration stems from the inherent mechanism of the adaptive penalty component, which filters low-probability tokens in the distribution tail—effectively limiting the diversity of generated content rather than specifically

Table 4: Experimental results on the three POPE subsets derived from MSCOCO with LLaVA-1.5 (7B) and Qwen-VL (7B).

| Model | LLaVA-1.5 | | | | | | | | | | | |
|---|---|---|---|---|---|---|---|---|---|---|---|---|
| Setting | *Random* | | | | *Popular* | | | | *Adversarial* | | | |
| Decoding | Acc | Precision | Recall | F1 | Acc | Precision | Recall | F1 | Acc | Precision | Recall | F1 |
| Regular | 83.2 | 91.7 | 73.0 | 81.2 | 81.8 | 88.9 | 72.8 | 80.0 | 78.9 | 83.1 | 72.7 | 77.5 |
| VCD | 87.6 | 92.8 | 81.4 | 86.7 | 82.6 | 87.1 | 80.5 | 83.3 | 77.3 | 73.4 | 86.4 | 79.2 |
| +ACD | 86.5 | 92.4 | 82.6 | 86.3 | 83.4 | 88.9 | 81.2 | 83.5 | 77.2 | 74.8 | 87.1 | 79.6 |
| ICD | 87.5 | 87.2 | 80.9 | 83.3 | 83.2 | 83.5 | 79.7 | 83.9 | 79.1 | 72.4 | 86.5 | 80.4 |
| +ACD | 87.1 | 87.5 | 82.5 | 83.4 | 83.5 | 84.2 | 80.9 | 84.0 | 78.8 | 73.2 | 86.6 | 80.8 |
| SID | 86.9 | 90.0 | 82.2 | 86.8 | 83.7 | 87.9 | 81.2 | 84.1 | 80.1 | 72.4 | 87.6 | 79.6 |
| +ACD | 85.2 | 89.4 | 80.8 | 84.8 | 83.6 | 86.9 | 84.3 | 85.6 | 81.7 | 73.6 | 88.8 | 81.2 |
| Model | Qwen-VL | | | | | | | | | | | |
| Setting | *Random* | | | | *Popular* | | | | *Adversarial* | | | |
| Decoding | Acc | Precision | Recall | F1 | Acc | Precision | Recall | F1 | Acc | Precision | Recall | F1 |
| Regular | 84.4 | 95.4 | 72.5 | 82.4 | 84.1 | 94.3 | 72.6 | 82.0 | 82.2 | 89.9 | 72.6 | 80.3 |
| VCD | 88.6 | 94.6 | 81.9 | 87.8 | 87.1 | 91.4 | 81.8 | 86.4 | 84.2 | 85.8 | 82.0 | 83.9 |
| +ACD | 87.5 | 93.5 | 81.0 | 85.8 | 86.8 | 92.0 | 80.6 | 85.3 | 81.9 | 85.5 | 80.4 | 83.1 |
| ICD | 88.0 | 94.3 | 80.8 | 85.4 | 85.5 | 90.8 | 79.5 | 84.0 | 81.6 | 84.3 | 78.5 | 81.5 |
| +ACD | 87.2 | 93.0 | 81.5 | 85.7 | 85.0 | 90.5 | 79.8 | 83.7 | 80.9 | 84.0 | 77.3 | 81.1 |
| SID | 87.3 | 94.0 | 80.4 | 84.9 | 85.2 | 89.9 | 78.7 | 83.3 | 80.3 | 82.8 | 76.9 | 79.5 |
| +ACD | 86.6 | 92.4 | 79.8 | 83.8 | 84.6 | 88.3 | 77.9 | 82.3 | 79.7 | 82.0 | 75.5 | 78.6 |

Table 5: Evaluation Results of Coherence Across Different Decoding Strategies.

| Method | baseline | VCD | +ACD |
|---|---|---|---|
| CIDEr | 0.266 | 0.231 | **0.266** |
| BLEU-4 | 0.094 | 0.054 | **0.096** |
| METEOR | 0.360 | 0.301 | **0.361** |
| ROUGE-L | 0.365 | 0.324 | **0.368** |

Table 6: Impact of Penalty Terms and ASS on VCD.

| Method | $Chair_s \downarrow$ | $Chair_i \downarrow$ | $Recall \uparrow$ |
|---|---|---|---|
| Regular | 54.1 | 18.5 | 73.4 |
| VCD | 51.8 | 16.2 | 76.8 |
| VCD W/O Pen. | 53.4 | 17.6 | 74.9 |
| **+ACD W/O Pen.** | **51.0** | **14.3** | **78.3** |
| +ACD W/O ASS | 52.1 | 15.6 | 77.4 |

Table 7: Ablation Study of $\alpha$ in ACD.

| $\alpha$ | $Chair_s \downarrow$ | $Chair_i \downarrow$ | $Recall \uparrow$ |
|---|---|---|---|
| 0.05 | 53.1 | 16.5 | 77.4 |
| 0.1 | 52.5 | 15.2 | 77.6 |
| 0.15 | **51.0** | **14.3** | **78.3** |
| 0.2 | 51.8 | 14.8 | 77.4 |
| 0.25 | 52.1 | 15.6 | 77.0 |

targeting hallucination reduction. Our proposed ACD methodology maintains robust hallucination mitigation capabilities without relying on an adaptive penalty term, thereby preserving generation quality while avoiding the diversity constraints imposed by traditional CD approaches. Concurrently, we validate the efficacy of the Adaptive Subtraction Strategy (ASS). Empirical evidence demonstrates that performance metrics deteriorate significantly when the ASS mechanism is disabled, as illustrated in Table 6. These findings underscore the critical role of adaptive subtraction in precisely suppressing hallucinatory components while preserving the model's factual generation capabilities.

**Effect of $\alpha$ in ACD.** To rigorously assess the impact of the amplification coefficient $\alpha$ in Equation 8—which modulates the contrastive intensity between distributions derived from pristine and perturbed visual inputs—we conducted a comprehensive ablation study across multiple $\alpha$ values according to the formulation in Equation 3. As evidenced in Table 7, our experiments reveal that $\alpha = 0.15$ yields optimal performance, establishing a critical hyperparameter threshold for effectively balancing contrastive signal strength while maintaining generation quality.

## 5 CONCLUSION

This paper addresses the challenge of balancing hallucination mitigation with output quality in LVLMs. We identified how existing Contrastive Decoding approaches compromise generation coherence through logits-layer operations that necessitate restrictive penalty mechanisms. Our Attention Contrastive Decoding framework shifts contrastive operations to the attention layer, where inherent smoothness promotes coherent generation, while the Adaptive Subtraction Strategy dynamically suppresses hallucination-prone attention patterns. Evaluations confirm ACD produces more coherent and factually accurate outputs without sacrificing diversity, resolving the quality-accuracy trade-off of prior approaches. By addressing hallucinations at their attentional source rather than through post-hoc adjustment, ACD enhances LVLM reliability for critical real-world applications.

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

## A  APPENDIX OVERVIEW

This appendix provides comprehensive supplementary information that complements the main manuscript, elaborating on methodological details, evaluation frameworks, and experimental analyses. Each section offers in-depth exploration of specific aspects of our Attention Contrastive Decoding (ACD) approach:

- **More Related Work (Section B):** Presents an extensive review of Large Vision-Language Models (LVLMs), their architectural components, applications, and the persistent challenge of hallucination phenomena.

- **Decoding Procedure (Section C):** Delineates the algorithmic implementation of ACD, elucidating how dual attention caches enable fine-grained intervention at the attention layer to mitigate hallucinations while preserving linguistic coherence.

- **Benchmarks and Evaluation Metric Details (Section D):** Explicates our comprehensive evaluation framework encompassing four distinct datasets (CHAIR, LLaVA-Bench, POPE, MME) with their corresponding metrics for systematic assessment of hallucination mitigation.

- **The Use of Large Language Models (Section E):** Clarifies the auxiliary role of LLMs in enhancing linguistic accuracy and coherence, while emphasizing that core research components were executed independently by the authors.

- **Ethics Statement (Section F):** Articulates our commitment to rigorous ethical standards, data anonymization protocols, and the societal implications of improved LVLM reliability in critical domains.

- **Reproducibility Statement (Section G):** Documents our adherence to reproducibility principles, including public release of code and comprehensive experimental configurations.

- **Comparison of Inference Speeds (Section H):** Presents benchmarking results demonstrating ACD's computational efficiency relative to baseline and VCD methods.

- **MME Full Set Results (Section I):** Reports comprehensive evaluations across the complete MME benchmark suite, highlighting ACD's consistent enhancement of perception, recognition, and complex reasoning capabilities.

- **Effect of VCD when LVLMs Scale Up (Section J):** Examines the scalability of our approach across different parameter configurations, demonstrating that ACD's benefits amplify with larger model capacities.

- **Effect of Different Sampling Strategies (Section K):** Analyzes how various decoding approaches affect the balance between output determinism and diversity, with empirical evidence of ACD's consistent hallucination mitigation across all strategies.

- **The Preprocessing Method of ACD (Section L):** Investigates normalization techniques for addressing attention scale disparities, with empirical validation of Temperature-scaled normalization's superiority.

- **Case Studies (Section M):** Presents illustrative examples of hallucination corrections achieved by our method, visually demonstrating improved coherence and factual accuracy.

- **Prompt for GPT-5 Evaluation (Section N):** Details the evaluation methodology employing GPT-5 as an automatic assessor for open-ended generation quality.

## B  MORE RELATED WORK

### B.1  LARGE VISION-LANGUAGE MODELS

Large Vision-Language Models (LVLMs)Liu et al. (2023b;c; 2024a); Zhu et al. (2023); Bai et al. (2023); Li et al. (2023a); Driess et al. (2023); Achiam et al. (2023); Chowdhery et al. (2023); Alayrac et al. (2022) have emerged as a pivotal research direction, effectively bridging computer vision and natural language processing paradigms. These models demonstrate exceptional capabilities across diverse multimodal tasks, including image captioningLin et al. (2014); Li et al. (2023a), visual

question answering (VQA)Liu et al. (2023c;b; 2024a); Hudson & Manning (2019), and sophisticated multimodal reasoningLu et al. (2022); Alayrac et al. (2022); Achiam et al. (2023). Modern LVLM architectures typically integrate visual encoders Tong et al. (2024) with feature projection modules Liu et al. (2023c; 2024a); Zhu et al. (2023) that interface with large language models (LLMs) Radford et al. (2018); Devlin et al. (2019); Brown et al. (2020); Raffel et al. (2020); Stiennon et al. (2020); Chowdhery et al. (2023); Thoppilan et al. (2022); Achiam et al. (2023), creating a unified embedding space where visual and textual representations converge to enable sophisticated cross-modal understanding. Despite significant advances in LVLMs, hallucination phenomena Li et al. (2023b); Fu et al. (2023b); Yue et al. (2024)—wherein models generate content fundamentally inconsistent with visual inputs—remain a critical limitation. This work addresses the challenge of hallucination mitigation in contemporary LVLMs, aiming to enhance their reliability and expand their applicability across diverse domains.

## C  DECODING PROCEDURE

During practical decoding, the ACD method maintains dual attention caches: an original cache and a hallucination cache. The specific algorithm is as follows:

---

**Algorithm 1** Generating Text with ACD Strategy

---

1: **Input:** Image $v$, text query $x$, hallucination induction method
2: **Output:** Generated text $y$
3: Initialize $y$ as an empty sequence
4: **for** $t = 1, 2, \ldots, T$ **do**
5:     Compute $q_t = \text{Query}(v, x, y_{<t})$
6:     Compute the clean attention $A_t^{\text{clean}} = \text{Softmax}(q_t K_{\leq t}^{\text{clean}\top}/\sqrt{d})$
7:     Compute the noisy attention $A_t^{\text{noisy}} = \text{Softmax}(q_t K_{\leq t}^{\text{noisy}\top}/\sqrt{d})$
8:     Generate the mask $M_t = I(A_t^{\text{clean}} - A_t^{\text{noisy}} < 0)$
9:     Apply the ASS strategy: $A_t^{\text{ACD}} = (1 + \alpha)A_t^{\text{clean}} - \alpha\, A_t^{\text{noisy}} \odot M_t$
10:     Use $A_t^{\text{ACD}}$ to compute the output distribution $p_\theta(y_t|v, x, y_{<t})$
11:     Sample to obtain the next token $y_t$
12:     Update the attention caches $K_{\leq t+1}^{\text{clean}}$ and $K_{\leq t+1}^{\text{noisy}}$
13: **end for**
14: **Return** $y$

---

At each decoding step, ACD computes attention distributions for both original and hallucination-inducing inputs, subsequently applying the ASS strategy to identify and suppress attention allocations potentially conducive to hallucination generation. This fine-grained intervention at the attention layer enables the model to maintain linguistic coherence and diversity while effectively mitigating hallucinations. In contrast to traditional CD methods, ACD eliminates the necessity for $\mathcal{V}_{\text{head}}$ penalty mechanisms as formulated in Equation 5 to filter low-probability tokens, as its attention-layer intervention inherently prevents invalid token generation. This characteristic allows ACD to reduce hallucinations while preserving content quality and diversity. The fundamental advantage of ACD lies in its exploitation of the inherent smoothness and structural continuity of the Transformer attention mechanism, enabling the model to precisely localize and suppress hallucination sources while maintaining generation quality. This approach provides a more reliable decoding solution for practical LVLM applications, effectively balancing hallucination mitigation with output quality preservation.

## D  BENCHMARKS AND EVALUATION METRIC DETAILS

This section delineates the comprehensive evaluation framework employed in our experimental analysis, encompassing four distinct benchmark datasets and their corresponding evaluation protocols designed to systematically assess hallucination mitigation efficacy across varied visual reasoning paradigms.

### D.1 CHAIR

CHAIR Yue et al. (2024) (Caption Hallucination Assessment with Image Relevance) quantifies object hallucination severity in generated image descriptions. Unlike conventional metrics that evaluate image descriptions through lexical or syntactic similarity, CHAIR directly leverages image annotation metadata to identify hallucination instances, thereby measuring semantic consistency between generated descriptions and actual visual content with greater precision. The CHAIR framework comprises two complementary metrics:

- $CHAIR_s$ (sentence-level): Measures the proportion of generated sentences containing at least one hallucinated object

- $CHAIR_i$ (instance-level): Quantifies the ratio of hallucinated object mentions to total object mentions across all generated descriptions

Additionally, recall serves as a complementary metric to evaluate semantic comprehensiveness, measuring the proportion of ground-truth objects successfully captured in the generated descriptions. This methodological approach circumvents potential biases inherent in purely linguistic matching techniques, offering enhanced sensitivity in hallucination detection while revealing semantic fidelity deficiencies that might remain undetected through traditional evaluation frameworks.

$$CHAIR_s = \frac{|\{\text{Captions with hallucinated objects}\}|}{|\{\text{All captions}\}|} \tag{10}$$

$$CHAIR_i = \frac{|\{\text{Hallucinated objects}\}|}{|\{\text{All mentioned objects}\}|} \tag{11}$$

$$Recall = \frac{|\{\text{Accurate objects}\}|}{|\{\text{Ground-truth objects}\}|} \tag{12}$$

### D.2 LLaVA-Bench (In-the-Wild)

LLaVA-Bench (In-the-Wild) Liu et al. (2023b) encompasses diverse challenging visual scenarios including indoor/outdoor environments, internet memes, paintings, sketches, and various artistic or abstract imagery typically underrepresented in standard training distributions. This benchmark deliberately introduces distributional shifts to assess model generalization capabilities across challenging visual contexts. The dataset comprises 24 distinct images with 60 corresponding questions, each image accompanied by meticulously crafted comprehensive descriptions and multiple targeted questions. Following an instruction-following evaluation paradigm, models receive visual inputs paired with natural language instructions requiring appropriate responses. The assessment protocol incorporates questions spanning fundamental visual comprehension to complex inferential reasoning. Given the absence of standardized ground truth evaluations, we employ GPT-5 OpenAI (2025), currently the most advanced Large Vision-Language Model available, as an automatic evaluator for generated descriptions. Through carefully engineered prompting (illustrated in Figure 1), the evaluation assesses model outputs along two critical dimensions: (1) Accuracy—measuring semantic consistency between generated descriptions and visual content; and (2) Detailedness—quantifying the richness and comprehensiveness of visual details captured in model-generated descriptions. This automated evaluation methodology enables consistent assessment of model performance across challenging out-of-distribution visual scenarios.

### D.3 POPE

POPE Li et al. (2023b) (Polling-based Object Presence Evaluation) formalizes the assessment of object hallucination as a binary classification task by prompting LVLMs with yes/no questions regarding object presence in images. Each evaluation sample consists of a triplet containing an image, corresponding question, and ground-truth answer. The framework constructs both positive and negative question instances with varying difficulty levels through three distinct sampling strategies: Random sampling—selecting arbitrarily from objects absent in the image; Popular sampling—selecting

the top $\frac{l}{2}$ most frequent objects across the dataset that are absent from the current image; and Adversarial sampling—selecting the top $k$ absent objects ranked by co-occurrence frequency with objects present in the image. Our POPE evaluation dataset incorporates over 20,000 question-answer pairs constructed from MSCOCO Lin et al. (2014), A-OKVQA Schwenk et al. (2022), and GQA Hudson & Manning (2019) datasets. Within the POPE evaluation framework, model responses are constrained to binary "Yes" or "No" outputs, effectively transforming object hallucination detection into a standard binary classification problem. Consequently, we employ established classification metrics—Accuracy, Precision, Recall, and F1-score—to quantitatively assess model performance. These metrics are formally defined as:

$$\text{Accuracy} = \frac{TP + TN}{TP + TN + FP + FN}, \tag{13}$$

$$\text{Precision} = \frac{TP}{TP + FP}, \tag{14}$$

$$\text{Recall} = \frac{TP}{TP + FN}, \tag{15}$$

$$\text{F1-score} = \frac{2 \cdot \text{Precision} \cdot \text{Recall}}{\text{Precision} + \text{Recall}}, \tag{16}$$

where $TP$, $TN$, $FP$, and $FN$ denote true positive, true negative, false positive, and false negative classifications, respectively. $Accuracy$ quantifies the proportion of correct predictions across all instances; $Precision$ measures the model's ability to avoid false positives when identifying present objects; $Recall$ assesses the model's capacity to identify all actually present objects; and $F1-score$ provides a harmonic mean of Precision and Recall, offering a comprehensive performance metric that balances both dimensions of classification quality.

### D.4 MME

MME Fu et al. (2023a) (Multimodal Model Evaluation) constitutes a comprehensive benchmark for multimodal large language model assessment, encompassing 14 distinct subtasks categorized into perceptual domains (object existence, quantification, color identification, spatial relationships) and cognitive domains (commonsense reasoning, numerical computation, text translation). To facilitate quantitative assessment, MME employs a standardized binary response format analogous to POPE, requiring "Yes/No" responses. A distinctive characteristic of this framework is its complementary instruction design—each visual input is paired with dual instructions, one requiring an affirmative response and the other a negative response based on ground truth. For evaluation metrics, MME implements an $accuracy$-based assessment methodology similar to POPE, calculating performance based on correct classification across individual instructions.

## E    THE USE OF LARGE LANGUAGE MODELS (LLMS)

Large Language Models (LLMs) served an auxiliary function in this work, enhancing linguistic accuracy, fluency, and coherence. These models assisted in identifying and rectifying grammatical errors, providing more precise lexical selections, and optimizing structural organization and logical progression to ensure greater clarity and readability. It is important to note that LLMs were not utilized for retrieval and discovery processes (e.g., identifying related work) or research ideation generation. These critical components were executed independently by the authors without computational linguistic assistance.

## F    ETHICS STATEMENT

The proposed Attention Contrastive Decoding (ACD) methodology, designed to enhance reliability and coherence in Large Vision-Language Models (LVLMs), adheres to rigorous ethical standards

Table 8: A comparison of inference speed and GPU memory usage for different methods applied to the LLaVA-v1.5 model on POPE subset benchmark.

| Method | Total Time | GPU-Memory | Latency/Example |
|--------|-----------|------------|-----------------|
| baseline | 13:46 | 14.7G | 0.276s |
| VCD | 26:51 | 15.8G | 0.537s |
| +ACD | **24:37** | **15.3G** | **0.493s** |

throughout all experimental procedures. Our research utilized publicly accessible datasets with appropriate anonymization protocols in compliance with established data ethics guidelines. We implemented deliberate measures to mitigate model bias and ensure equitable performance across diverse contexts. To facilitate scientific transparency and reproducibility, we have released our experimental codebase publicly, enabling independent verification and extension of our findings by the research community. The societal implications of this work are potentially significant, as ACD demonstrably reduces hallucination generation, thereby enhancing the accuracy and reliability of automated content generation. Such improvements could yield substantial benefits in critical domains including healthcare diagnostics and educational applications where factual precision is paramount.

## G    REPRODUCIBILITY STATEMENT

This research strictly adheres to reproducibility principles fundamental to scientific advancement. All experiments were conducted using publicly accessible benchmark datasets, with comprehensive implementation details of the proposed Attention Contrastive Decoding (ACD) methodology provided. Our codebase, including configuration files and execution scripts with detailed reproduction instructions, is publicly available in our open-source repository (`https://anonymous.4open.science/r/ACD-00C6`). To facilitate thorough replication, we have meticulously documented all experimental configurations, hyperparameters, optimization protocols, and computational resource specifications. This documentation encompasses training environments, inference procedures, and evaluation methodologies, thereby enabling independent verification of our empirical findings. This transparency not only supports validation of our reported results but also establishes a foundation for future extensions and applications of the ACD approach across various multimodal generation tasks.

## H    COMPARISON OF INFERENCE SPEEDS

To evaluate the computational efficiency of our method, we benchmarked the LLaVA-v1.5 model on the POPE subset of MSCOCO random benchmark. All experiments were conducted on a server equipped with a single NVIDIA RTX 3090 24GB GPU. As demonstrated in Table 8, the CD method approximately doubles the runtime compared to conventional decoding while maintaining similar memory utilization. Compared to VCD, our proposed ACD framework demonstrates enhanced hallucination mitigation capabilities while maintaining computational efficiency. ACD achieves reduced memory footprint and inference latency by computing attention exclusively for hallucination-prone inputs, thereby avoiding the computation of hidden state outputs that VCD necessitates. This targeted computational approach enables our method to achieve superior performance without introducing additional computational or memory overhead, resulting in a more efficient solution for hallucination mitigation in vision-language models.

## I    MME FULL SET RESULTS

As illustrated in Figure 3, we conducted comprehensive evaluations using the complete MME benchmark suite to assess the impact on general capabilities of Large Vision-Language Models (LVLMs). We present results from LLaVA-1.5 as representative among models exhibiting comparable performance trajectories. Notably, our ACD methodology demonstrates consistent enhancement across perception and recognition tasks when compared to the original VCD approach. The improvements are particularly pronounced in complex reasoning domains, with substantial gains observed in Common Sense Reasoning, Optical Character Recognition (OCR), and Text Translation tasks. These

Table 9: An ablation study of different sampling strategies.

| Sampling Strategy | VCD w. ACD | $Chair_s\downarrow$ | $Chair_i\downarrow$ | $Recall\uparrow$ |
|---|---|---|---|---|
| Greedy | No | 51.6 | 15.3 | 77.4 |
| | Yes | **50.5** | **14.1** | **79.2** |
| Sampling | No | 51.8 | 16.2 | 76.8 |
| | Yes | **51.0** | **14.3** | **78.3** |
| Top P | No | 51.8 | 16.3 | 76.5 |
| | Yes | **51.0** | **14.3** | **78.4** |
| Top K | No | 52.4 | 17.6 | 75.9 |
| | Yes | **51.6** | **15.2** | **77.1** |
| Top K+Temperature 0.5 | No | 51.8 | 16.0 | 77.1 |
| | Yes | **50.9** | **14.3** | **79.1** |

Table 10: The preprocessing method of ACD.

| Method | $Chair_s\downarrow$ | $Chair_i\downarrow$ | $Recall\uparrow$ |
|---|---|---|---|
| Direct Subtraction | 52.7 | 16.2 | 77.8 |
| Layer-scaled | 51.6 | 14.7 | 78.9 |
| Temperature-scaled | **51.0** | **14.3** | **78.3** |

results suggest that our attention-based contrastive mechanism not only preserves essential visual understanding capabilities but significantly enhances performance on cognitively demanding tasks while effectively mitigating hallucination tendencies.

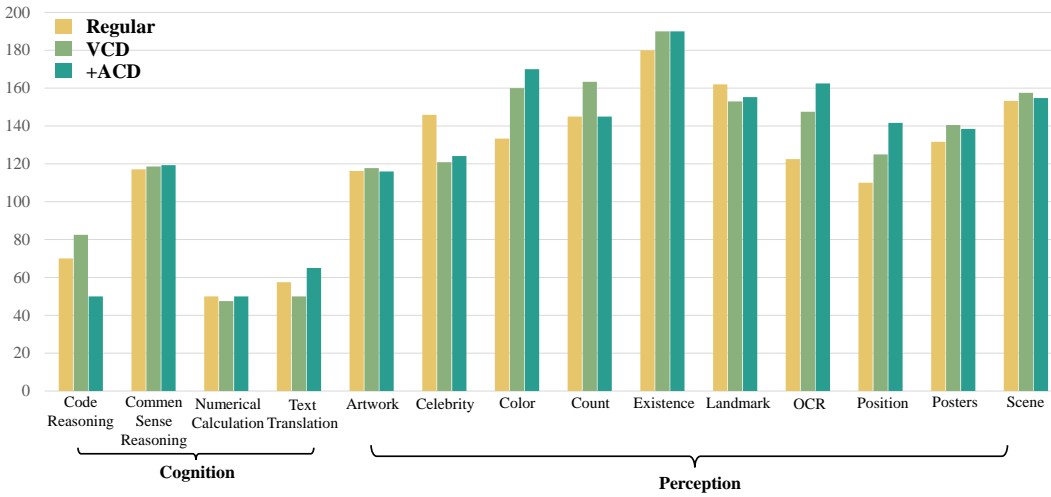

Figure 3: MME full set results on LLaVA-1.5

## J  EFFECT OF VCD WHEN LVLMS SCALE UP

To investigate the scalability and generalizability of our proposed ACD approach, we conducted comprehensive evaluations using LVLMs of different parameter sizes. Table 11 presents the comparative performance of the standard VCD against our proposed ACD method on the LLaVA-1.5 architecture with both 7B and 13B parameter configurations. The experimental results demonstrate that ACD consistently outperforms the baseline VCD approach across all evaluation metrics and model scales. For the 7B parameter configuration, ACD yields substantial improvements in accuracy across all categories, with particularly notable gains in conversational (+2.16) and detail-oriented (+3.00) tasks. Similarly, for the 13B parameter configuration, we observe even more pronounced enhancements, with accuracy improvements of +2.40 in conversational tasks and +2.98 in detail-

Table 11: Experimental evaluation of GPT-5 was conducted on the LLaVA-Bench-Wild dataset using LLaVA-1.5 architectures with 7B and 13B parameter configurations.

| Model | Decoding | Conv | | Detail | | Complex | | Total | |
|-------|----------|------|--------|------|--------|------|--------|------|--------|
| | | Acc. | Detail | Acc. | Detail | Acc. | Detail | Acc. | Detail |
| LLaVA-1.5(7B) | VCD | 4.67 | 3.67 | 4.50 | 5.67 | 3.88 | 4.88 | 4.30 | 4.75 |
| | **+ACD** | **6.83** | **3.71** | **7.50** | **7.33** | **3.97** | **5.21** | **5.89** | **5.40** |
| LLaVA-1.5(13B) | VCD | 5.32 | 4.50 | 5.25 | 6.10 | 4.92 | 5.30 | 5.10 | 5.85 |
| | **+ACD** | **7.72** | **4.90** | **8.23** | **7.88** | **5.00** | **5.80** | **6.70** | **6.25** |

oriented scenarios. Importantly, the performance gap between VCD and ACD remains consistent or even widens as model size increases, suggesting that the benefits of our approach are not diminished but rather amplified with larger model capacities. This pattern is particularly evident in the total accuracy metrics, where ACD improves performance by +1.59 and +1.60 percentage points for the 7B and 13B models, respectively. These findings strongly indicate that ACD significantly enhances performance across all model configurations, further substantiating its robustness independent of model scale. The consistent improvements across different parameter sizes demonstrate that our proposed method addresses fundamental limitations in vision-language reasoning that persist even as models grow in capacity, highlighting the complementary nature of our approach to architectural scaling.

## K    EFFECT OF DIFFERENT SAMPLING STRATEGIES

In the text generation process of large vision-language models (LVLMs), decoding strategies critically determine the balance between output determinism and diversity. Greedy search, which invariably selects the token with maximum conditional probability at each step, produces consistent outputs but suffers from limited diversity. To introduce stochasticity, direct sampling draws from the complete probability distribution, enhancing diversity but remaining susceptible to interference from low-probability noise tokens. To achieve an optimal equilibrium, top-k sampling preserves only the k highest-probability candidate tokens for sampling, thereby mitigating the influence of extremely low-probability terms. Extending this approach, top-p sampling (nucleus sampling) dynamically determines the candidate set size by establishing a cumulative probability threshold p, allowing the sampling scope to adapt to the probability distribution's steepness and thus more flexibly balancing coherence and diversity. Building upon these methods, combining top-k sampling with temperature scaling amplifies the advantage of high-probability tokens by rescaling the distribution, maintaining top-k constraints while simultaneously enhancing generation stability and determinism. Beyond direct sampling, we conducted an ablation study employing various decoding strategies on the CHAIR dataset using LLaVA-1.5. The results presented in Table 9 demonstrate that ACD consistently mitigates hallucinations across all implemented decoding strategies.

## L    THE PREPROCESSING METHOD OF ACD

For our ACD method, which employs an attention-level contrastive adjustment mechanism, potential scale disparities in attention necessitate appropriate normalization preprocessing. We investigated multiple normalization techniques, specifically Layer-scaled and Temperature-scaled approaches. Empirical evaluation using LLaVA-1.5 on the CHAIR dataset reveals that, as demonstrated in Table 10, the Temperature-scaled method yields superior performance, indicating its enhanced capacity to equilibrate attention scales of varying magnitudes, thereby effectively mitigating hallucination generation. These findings suggest that Temperature-scaled normalization offers significant advantages in improving model generation stability and semantic coherence, particularly when processing complex vision-language multimodal tasks where precise control of attention distribution optimizes generation quality.

## M  CASE STUDIES

Case studies on the chair and LLaVA-bench are presented to illustrate the effectiveness of our approach. Figures4,5, 6, and 7 provides instances of hallucination corrections achieved by our method ACD. From a perceptual standpoint, our method generates more coherent and diverse content while offering supplemental examples of the enhancements brought by our proposed ACD in bolstering the general perception and recognition abilities of LVLMs.

## N  PROMPT FOR GPT-5 EVALUATION

To evaluate open-ended generation, we employ GPT-5 to assess the accuracy and detailedness of LVLMs' responses. The specific configurations are delineated in Figure 8.

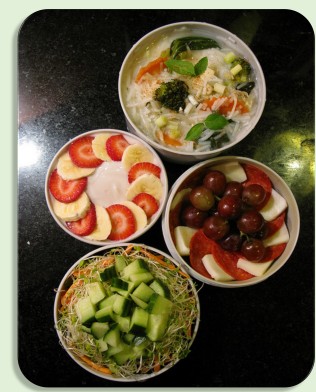

### Describe this photo in detail

**GT：** The image features a dining table with four bowls filled with various types of food. The bowls are placed in a row, with each bowl containing different types of food. The first bowl on the left contains a mix of fruits, including bananas and strawberries. The second bowl, located in the middle, has a soup with broccoli and carrots. The third bowl on the right is filled with grapes, and the fourth bowl on the far right has a salad. In addition to the bowls, there are a few carrots scattered around the table, with some placed near the middle bowl and others near the rightmost bowl. The table setting creates an inviting atmosphere for a meal or gathering.

**Baseline：** The image features a dining table with **three bowls** filled with a variety of fruits and vegetables. The bowls are placed in a line, with each bowl containing different types of food. The first bowl on the left contains a mix of fruits **and vegetables, including bananas, apples, and carrots**. The second bowl, in the middle, is filled with a combination of **grapes** and broccoli. The third bowl on the right is filled with a mix of fruits **and vegetables, such as strawberries, carrots, and apples. In addition to the bowls, there is a spoon placed on the table, likely for serving the food.** The arrangement of the bowls creates an inviting and healthy display of food options.

— — — — — — — — — — — — — — — — — — — — — — — — — — — — — — — — — — — — — — — — —

**VCD：** The image features a dining table with **three bowls** filled with various fruits and vegetables. The bowls are placed in a row, with each bowl containing different types of food. The fruits and vegetables in the bowls include **apples**, bananas, and carrots. In addition to the bowls, there are a few carrots **scattered around the table, some of which are placed near the edge of the table**. The table setting creates an inviting and healthy atmosphere, perfect for enjoying a meal with friends or family.

**+ACD ：** The image displays a dining table with four distinct bowls placed on it, each holding various vegetables and fruits. One of the bowls contains bananas and strawberries, while another has sliced bananas in cream. There is also a bowl with broccoli and carrots, along with **two others** that contain grape and apple slices. The table has an enticing assortment of foods that showcase a balance of healthy ingredients.

Figure 4: More examples from CHAIR of our proposed ACD for hallucination corrections. Hallucinated parts from LVLM's decoding are highlighted in red.

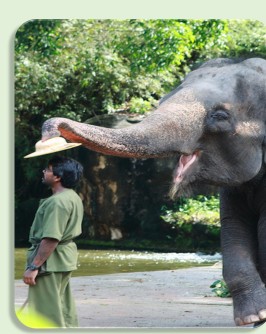

## Describe this photo in detail

**GT**: The image features a man standing next to a large elephant, both of them in a natural setting. The man is holding a hat in his hand, and the elephant is reaching out to take the hat from him. The elephant's trunk is extended towards the man, creating a unique and captivating moment between the two. The man appears to be enjoying the interaction with the elephant, and the scene captures the essence of a close encounter with these majestic creatures.

**Baseline**: The image features a man standing next to a large elephant, both of them in an outdoor setting. The man is wearing a hat and appears to be feeding the elephant, **while the elephant is eating from a hat held by the man.** The elephant is positioned on the right side of the man, and they are both standing close to each other. In the background, **there is a bench located on the left side of the scene,** and a potted plant can be seen in the middle of the image. The overall atmosphere of the scene suggests a friendly interaction between the man and the elephant.

**VCD**: The image captures an intriguing scene featuring an old **elephant walking into the water next to a man. The elephant is putting its trunk into its mouth, walking through the water. The water appears to be deep**, up to the elephant's sides. **There are at least two people present in the scene one near the far left of the image and another one near the middle right side.** A hat can be spotted close to the top left corner of the frame, **while an ironing board is positioned in the middle foreground, indicating that someone may have recently been ironing nearby. A backpack rests near the middle of the image and a spoon can be seen near the elephant, potentially being used for feeding.**

**+ACD**: The image features a man standing next to a large elephant, with the elephant's trunk reaching up to the man's hat. The man is wearing a green shirt and appears to be feeding the elephant. The elephant is positioned on the right side of the man, occupying a significant portion of the scene. In the background, **there is a bench located on the left side of the image**, and a potted plant can be seen in the middle of the scene. The overall atmosphere suggests a close encounter with the majestic animal..

Figure 5: More examples from CHAIR of our proposed ACD for hallucination corrections. Hallucinated parts from LVLM's decoding are highlighted in red.

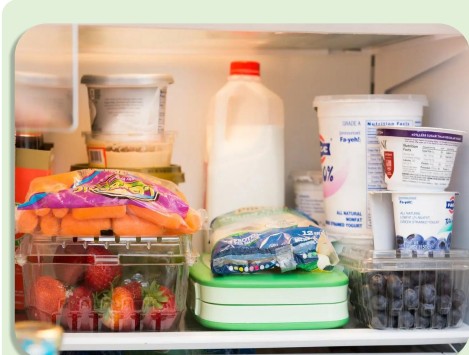

### Describe this photo in detail

**GPT-5:** The photo shows the contents of a well-stocked refrigerator shelf viewed straight on. Center back: a white plastic gallon of milk with a red cap. Right side: several containers of yogurt. There's a tall white tub behind, a smaller cup stacked on a yogurt cup below it, and nutrition labels visible. In front-right, a clear clamshell of blueberries sits on the shelf. Left side: a clear clamshell of strawberries on the lower shelf area, with bright red berries and green tops visible. Resting on top of that clamshell is an opened plastic bag of baby carrots with a purple-and-orange label. Middle foreground: a light-green, stackable lunchbox or food container with rounded corners; on top of it lies a blue package, possibly tortillas or flatbreads. Back left: a couple of lidded tubs, likely dips or spreads (one may be hummus), with another short container partly hidden behind them. Overall impression: fresh produce (strawberries, blueberries, carrots) and dairy (milk, Greek yogurts) organized in clear containers, with neutral white refrigerator walls and bright, even lighting.

**Baseline:** The image displays a well-stocked refrigerator filled with various food items. Inside the refrigerator, there are bottles, including milk, **juice**, and other beverages. **Some of the bottles are placed on the top shelf, while others are located on the middle and bottom shelves.** In addition to the bottles, the refrigerator also contains several fruits, such as strawberries and blueberries. **The strawberries are placed on the bottom shelf, while the blueberries are scattered throughout the refrigerator.** The refrigerator is also filled with carrots, **which are located on the middle shelf.** Overall, the refrigerator is well-organized and stocked with a variety of food items, making it an inviting and convenient space for meal preparation.

‒ ‒ ‒ ‒ ‒ ‒ ‒ ‒ ‒ ‒ ‒ ‒ ‒ ‒ ‒ ‒ ‒ ‒ ‒ ‒ ‒ ‒ ‒ ‒ ‒ ‒ ‒ ‒ ‒ ‒ ‒ ‒ ‒ ‒ ‒ ‒ ‒ ‒ ‒ ‒ ‒ ‒ ‒

**VCD:** The image displays an open refrigerator filled with a variety of food items. Inside the refrigerator, **there are multiple carrots placed in different sections, with some located near the top, middle, and bottom of the fridge.** A carton of yogurt can be seen on the right side of the refrigerator, while a bottle is positioned on the left side. In addition to the food items, **there are two strawberries** placed on the left side of the refrigerator, adding a touch of color to the scene. The refrigerator is well-stocked with various foods, making it a convenient place to store and access nutritious meals.

**+ACD:** The image displays an open refrigerator filled with various food items and beverages. Inside the refrigerator, there are multiple bottles, some of which are placed on the shelves, while others are standing upright. The bottles vary in size and shape, and they are arranged in different positions within the refrigerator. In addition to the bottles, the refrigerator also contains a few bowls, with one placed on the top shelf and another on the middle shelf. There are also a couple of carrots stored in the refrigerator, **one on the bottom shelf** and another on the middle shelf. The refrigerator is well-stocked with a variety of food items and beverages, making it a convenient place to store and access groceries.

Figure 6: More examples from LLaVA-Bench of our proposed ACD for hallucination corrections. Hallucinated parts from LVLM's decoding are highlighted in red.

### Describe this photo in detail

**GPT-5:** The photo shows a busy multilane highway at night, viewed from above. Two wide carriageways curve through the frame from bottom left toward the upper right, separated by a narrow, planted median. Traffic is heavy in both directions: dense streams of white headlights on the left side and red taillights on the right, creating a ribbon of bright, colorful dots and short light trails. Vehicles range from sedans and SUVs to a few buses and trucks, packed bumper-to-bumper in some stretches.Streetlights line the road on tall poles, casting warm pools of light onto the asphalt. On the far right edge is a darker strip of trees and utility poles, with a sidewalk and scattered pedestrians or cyclists near bus stops lit by small shelters. To the far left, an adjacent service road runs parallel with lighter traffic. The overall scene conveys the energy and congestion of an urban evening rush hour

**Baseline:** The image depicts a busy highway at night, filled with heavy traffic. The highway is surrounded by trees, adding to the scenery. There are numerous cars of various sizes and colors, **all moving in the same direction.** Some cars are closer to the foreground, while others are further away, creating a sense of depth in the scene. The traffic appears to be moving smoothly, despite the high volume of vehicles. The highway is well-lit, with streetlights illuminating the area, making it easier for drivers to navigate the busy roads.

------------------------------------------------

**VCD:** The image depicts a bustling highway filled with heavy traffic, including numerous cars and trucks. The highway is a busy four-lane road, with traffic in both directions. The traffic is dense, with cars and trucks occupying the lanes, and some vehicles are stopped or moving at a slow pace. In addition to the vehicles, there are several pedestrians visible in the scene, walking along the side of the highway or **crossing the road**. Traffic lights can be seen at various points along the highway, indicating the presence of intersections and ensuring the safe flow of traffic.

**+ACD:** The image depicts a busy highway at night, filled with a large number of cars traveling in both directions. The highway is surrounded by trees, adding to the scenery. The traffic is dense, with cars occupying almost every lane, and some vehicles are positioned closer to the center of the highway. There are multiple traffic lights visible in the scene, ensuring the smooth flow of traffic. Some of these traffic lights are located near the center of the highway, while others are positioned closer to the edges. The overall atmosphere of the image is bustling and filled with activity, as people commute through the city at night.

Figure 7: More examples from LLaVA-Bench of our proposed ACD for hallucination corrections. Hallucinated parts from LVLM's decoding are highlighted in red.

**Description:**
AI that scores image description accuracy and detailedness.

**Instructions:**
You are an AI designed to evaluate and score the performance of two AI assistants in describing a given image. Your primary focus is on the accuracy and detailedness of their descriptions. You will assess the accuracy by checking for hallucinations - any part of the description that is inconsistent with the image content. For detailedness, you will consider how rich the response is in necessary details, excluding any hallucinated parts. You will provide scores on a scale from 1 to 10 for each assistant separately, based on these criteria. After scoring, you will offer an explanation for your evaluation, ensuring it is free from bias and not influenced by the order of presentation of the responses.

Input format:

[Assistant 1]
{Response 1}
[End of Assistant 1]

[Assistant 2]
{Response 2}
[End of Assistant 2]

Output format:

Accuracy:
Scores of the two answers:
Reason:

Detailedness:
Scores of the two answers:
Reason:

Figure 8: The configuration to build an image-description evaluator with GPT-5.

