# OpenReview forum: "Attention Contrastive Decoding: Preserving Coherence While Mitigating Hallucinations in Large Vision-Language Models"
_ICLR.cc/2026/Conference — ICLR 2026 Conference Withdrawn Submission_

### Official Review · Reviewer_RMtV · 2025-10-15

**Soundness:** 2
**Presentation:** 2
**Contribution:** 3
**Rating:** 4
**Confidence:** 5

**Summary:**

The paper proposes Attention Contrastive Decoding (ACD), a training-free decoding strategy for mitigating hallucinations in LVLMs. Unlike existing CD methods that operate at the logits level and harm output coherence and diversity, ACD shifts the contrastive operation to the attention layer. It introduces an Adaptive Subtraction Strategy (ASS) that compares attention patterns between clean and hallucination-inducing inputs to selectively suppress hallucination-prone regions. Experiments show that ACD reduces hallucinations while preserving or even improving linguistic coherence and generation diversity.

**Strengths:**

1. Innovative Mechanism: By moving contrastive decoding from the volatile logits space to the smoother attention space, ACD better preserves language model fluency and avoids the coherence degradation seen in prior CD methods.

2. Effective and Penalty-Free: ACD achieves strong hallucination suppression without relying on restrictive token filtering or heuristic penalty thresholds, thereby maintaining output diversity and naturalness.

3. Strong Empirical Validation: The method is rigorously evaluated across four standard benchmarks and multiple LVLMs, outperforming many CD baselines in both factual accuracy and generation quality.

**Weaknesses:**

1. The Adaptive Subtraction Strategy (ASS) is presented as one of the key innovations of the paper. However, the ablation study for ASS is only conducted on the CHAIR benchmark and solely in conjunction with VCD. It remains unclear how much performance gain ASS provides on discriminative benchmarks such as POPE or MME, and when integrated with other baseline contrastive decoding methods.

2. Attention maps are widely recognized as effective indicators of a model’s reliance on visual inputs, and the proposed ASS leverages this intuition. Nevertheless, the paper lacks visualizations or qualitative analyses of the attention patterns before and after applying ASS. This omission undermines the interpretability of the method. As a result, the explanation for the mask M_t may not be sufficiently convincing without empirical or visual evidence supporting its alignment with hallucination-prone regions.

3. The chosen baselines appear somewhat outdated. Recent works such as Mixture of Decoding (MoD) and Cross-Image Contrastive Decoding (CICD) have demonstrated superior hallucination mitigation performance. It would strengthen the paper significantly if the authors evaluate ACD on top of these more advanced contrastive decoding frameworks to demonstrate its generalizability and improve performance under stronger baselines.

**Questions:**

See Weaknesses.

---

### Official Review · Reviewer_a2C6 · 2025-10-27

**Soundness:** 2
**Presentation:** 2
**Contribution:** 2
**Rating:** 4
**Confidence:** 3

**Summary:**

This paper introduces Attention Contrastive Decoding (ACD), a novel decoding strategy designed to mitigate hallucinations in large vision-language models (LVLMs) without sacrificing generation coherence or diversity  . Unlike conventional Contrastive Decoding (CD) methods that subtract hallucination-inducing logits distributions and thereby disrupt linguistic fluency, ACD performs contrastive operations at the attention layer, leveraging an Adaptive Subtraction Strategy (ASS) to identify and suppress hallucination-prone attention patterns. By comparing attention responses between original and noise-induced inputs, the method adaptively filters unreliable regions while preserving valid visual-linguistic dependencies. Extensive experiments on benchmarks such as CHAIR, POPE, MME, and LLaVA-Bench demonstrate that ACD substantially reduces hallucinations, improves factual accuracy, and maintains linguistic coherence without requiring penalty mechanisms or additional training.

**Strengths:**

1. Performing contrastive decoding at the attention level rather than on the output logits is an interesting idea.
2. Experiments conducted on multiple benchmarks demonstrate the effectiveness of the proposed method.
3. Eliminating the need for penalty mechanisms is an interesting exploration, with experimental results demonstrating the effectiveness of the proposed method in achieving this.

**Weaknesses:**

1. The motivation behind Eq. 9 in the paper is not clearly explained. Specifically, it is unclear why A_clean − A_noisy > 0 is interpreted as indicating semantic information loss in the corresponding region. What exactly does “semantic information loss” mean in this context? Furthermore, why does A_clean − A_noisy < 0 imply that the model is potentially precipitating hallucination generation? These points require clearer illustration and explanation; otherwise, it is difficult to assess the soundness of the proposed method.
2. Although the authors have conducted experiments on different LLMs (LLaVA and Qwen-VL), it is recommended to include results on more advanced models, such as Qwen3-VL.
3. This work focuses solely on multimodal LLMs, but the proposed method does not appear to include any design specifically tailored to multimodality. Therefore, it is recommended to further evaluate the generalization and applicability of ACD on text-only LLMs and corresponding benchmarks.
4. All experimental settings are based on integrating ACD into existing methods (e.g., VCD+ACD, ICD+ACD). Can the proposed ACD be used independently without relying on these existing approaches?
5. As shown in Table 7, the performance fluctuates noticeably (CHAIRs from 51.0 to 52.5) when $\alpha$ decreases from 0.15 to 0.1, indicating that the proposed method may not be very robust to hyperparameter settings.

**Questions:**

Please see the Weaknesses section.

---

### Official Review · Reviewer_BRD4 · 2025-11-01

**Soundness:** 2
**Presentation:** 2
**Contribution:** 2
**Rating:** 4
**Confidence:** 3

**Summary:**

This paper points out that although previous Contrastive Decoding (CD) methods effectively reduce hallucinations, their logit-level operations degrade the quality of language generation.
To address this issue, the authors perform CD at the attention layer instead of the logit layer and introduce the Adaptive Subtraction Strategy (ASS) to dynamically suppress attention.
Through this approach, the model improves coherence and mitigates hallucinations more effectively than previous CD methods.

**Strengths:**

- This paper is meaningful in that it raises the issue of coherence in the highly active field of Contrastive Decoding (CD) research.

- To address this problem, the authors apply CD at the attention layer, effectively mitigating hallucinations through this approach.

**Weaknesses:**

1. There is a lack of experiments on recent models. This paper conducts experiments using LLaVA 1.5 and Qwen-VL, but does not include evaluations on newer models such as Qwen2.5-VL or InternVL3. Experiments on these latest models appear necessary.

2. The clarity of writing is also insufficient. In the experimental section, the paper states that the ACD method integrates various hallucination-inducing scenarios, which is ambiguous. Most tables (e.g., Table 1 and Table 2\) use notations like VCD \+ ACD, making the structure unclear. Does this integration refer to the combination at the noise or input level (as in VCD, ICD, and SID)? If so, the paper should explain this point in greater detail and separate the tables or clarify their captions, since the current format gives the impression that ACD is stacked on top of existing CD methods.

**Questions:**

1. How were the hyperparameter settings for noise-related components determined? Did the authors follow the configurations from previous studies? Also, how was the hyperparameter α (alpha) chosen and applied — was it obtained directly from the test set, or determined through a separate validation process?
2. The method in this paper does not appear to be specific to images. (LVLM) Does this mean the method can be extended to LLMs as well? Additionally, I am curious whether the issues identified in the paper persist in pure LLMs (without visual input).
3. Is this method compatible with FlashAttention?Previous contrastive decoding methods are generally compatible with FlashAttention, but since the proposed method directly accesses the attention mechanism, I am wondering whether it can still operate with FlashAttention enabled. In Appendix Table 8, were the results for the baseline, VCD, and ACD all obtained without using FlashAttention? If not, could you clarify whether FlashAttention was used for methods that support it, and if so, please provide the runtime and GPU memory usage when FlashAttention is enabled?

Minor comment: In the method section, although Equation (8) includes the term ASS(Aclean, Anoisy), Section 3.3 does not provide a clear corresponding mathematical formulation, which reduces readability. Adding an explicit equation for ASS would significantly improve the paper’s clarity and overall comprehensibility.

---

### Official Review · Reviewer_oHTr · 2025-11-01

**Soundness:** 3
**Presentation:** 2
**Contribution:** 2
**Rating:** 2
**Confidence:** 5

**Summary:**

This paper introduces a new decoding strategy called Attention Contrastive Decoding (ACD) to address the trade-off between hallucination mitigation and output coherence in Large Vision-Language Models (LVLMs). Previous training-free contrastive decoding (CD) methods operate at the logits level, which can suppress hallucinations but often produce incoherent or repetitive text. To overcome this, ACD moves the contrastive operation to the attention layer, leveraging the smoother, context-aware properties of attention mechanisms. It also introduces an Adaptive Subtraction Strategy (ASS) that dynamically identifies and suppresses attention patterns associated with hallucinations while maintaining linguistic fluency. Experiments on benchmarks such as CHAIR, POPE, MME, and LLaVA-Bench-Wild show that ACD achieves lower hallucination rates and higher coherence scores compared to baseline CD methods, all without using penalty mechanisms or retraining. The results suggest that handling hallucination at the attention level yields more stable and faithful multimodal generation.

**Strengths:**

* The proposed method mitigates hallucinations while preserving decoder coherence by applying contrastive decoding directly on the attention map, rather than on the output logits.

* Experimental results show that the proposed approach is effective in balancing hallucination reduction and coherence preservation, especially when combined with existing contrastive decoding (CD) methods.

**Weaknesses:**

* There already exist training-free attention-masking methods that mitigate hallucinations by modifying the attention map [1-4]. However, the paper does not include comparisons with such approaches. Since the proposed method is only evaluated as an extension of existing CD methods, it would be beneficial to add additional baselines representing prior attention-based hallucination mitigation techniques.

* The experiments evaluate response coherence using CIDEr, BLEU, and other n-gram overlap metrics with ground truth references. These metrics are not ideal for measuring coherence in generated text [5-7].  To more accurately assess coherence, the paper should include human evaluation or, at minimum, LLM-based automatic evaluation to complement the existing metrics.

* The proposed approach computes an adaptive mask at each decoding step, which likely introduces latency overhead. A quantitative analysis of this additional computational cost would improve the completeness of the evaluation.

* Table 7 presents sensitivity results for hyperparameter changes, but this is not a rigorous ablation study. It would be helpful if the authors clarified the backbone model and base decoding method used in this experiment.

Minor:
The paper contains multiple formatting and citation inconsistencies. For example, the reference style does not follow the official formatting guidelines, and the terms LVLM and MLLM are used interchangeably (e.g., in Figure 1). It would be better to standardize terminology throughout the paper for clarity and consistency.

**References**

[1] Ye, Xubing, et al. "Atp-llava: Adaptive token pruning for large vision language models." Proceedings of the Computer Vision and Pattern Recognition Conference. 2025.

[2] An, Wenbin, et al. "Mitigating object hallucinations in large vision-language models with assembly of global and local attention." Proceedings of the Computer Vision and Pattern Recognition Conference. 2025.

[3] Tang, Feilong, et al. "Seeing Far and Clearly: Mitigating Hallucinations in MLLMs with Attention Causal Decoding." Proceedings of the Computer Vision and Pattern Recognition Conference. 2025.

[4] Gong, Xuan, et al. "DAMRO: Dive into the Attention Mechanism of LVLM to Reduce Object Hallucination." Proceedings of the 2024 Conference on Empirical Methods in Natural Language Processing. 2024.

[5] Cui, Yin, et al. "Learning to evaluate image captioning." Proceedings of the IEEE conference on computer vision and pattern recognition. 2018.

[6] Wang, Qingzhong, and Antoni B. Chan. "Describing like humans: on diversity in image captioning." Proceedings of the IEEE/CVF Conference on Computer Vision and Pattern Recognition. 2019.

[7] Scialom, Thomas, et al. "QuestEval: Summarization Asks for Fact-based Evaluation." Proceedings of the 2021 Conference on Empirical Methods in Natural Language Processing. Association for Computational Linguistics, 2021.

**Questions:**

See weakness

---

### Note · Authors · 2025-11-14

I have read and agree with the venue's withdrawal policy on behalf of myself and my co-authors.